# Atmospheric circulation and hydroclimate impacts of alternative warming scenarios for the Eocene

# Henrik Carlson<sup>1</sup> and Rodrigo Caballero<sup>1</sup>

<sup>1</sup>Department of Meteorology and Bolin Centre for Climate Research, Stockholm University *Correspondence to:* Henrik Carlson (henrik@misu.su.se)

Abstract. Recent work in modelling the warm climates of the Early Eocene shows that it is possible to obtain a reasonable global match between model surface temperature and proxy reconstructions, but only by using extremely high atmospheric  $CO_2$  concentrations or more modest  $CO_2$  levels complemented by a reduction in global cloud albedo. Understanding the mix of radiative forcing that gave rise to Eocene warmth has important implications for constraining Earth's climate sensitivity, but progress in this direction is hampered by the lack of direct proxy constraints on cloud properties. Here, we explore the potential for distinguishing among different radiative forcing scenarios via their impact on regional climate changes. We do this by comparing climate model simulations of two end-member scenarios: one in which the climate is warmed entirely by  $CO_2$  (which we refer to as the GHG scenario), and another in which it is warmed entirely by reduced cloud albedo (which we refer to as the "low  $CO_2$ -thin clouds" or LCTC scenario). The two simulations have almost identical global-mean surface

- 10 temperature and equator-to-pole temperature difference, but the LCTC scenario has ~11% greater global-mean precipitation than the GHG scenario. The LCTC scenario also has cooler midlatitude continents and warmer oceans than the GHG scenario, and a tropical climate which is significantly more El Niño-like. Extremely high warm-season temperatures in the subtropics are mitigated in the LCTC scenario, while cool season temperatures are lower at all latitudes. These changes appear large enough to motivate further, more detailed study using other climate models and a more realistic set of
- 15 modelling assumptions.

5

#### 1 Introduction

The Early Eocene ( $\sim$ 50 Ma) was characterized by very warm surface temperatures compared with the present day (Huber, 2008; Pagani et al., 2014). Considerable progress has been made recently in reconciling proxy temperature reconstructions with climate model simulations of this period: models can now capture both the reconstructed global-mean and equator-pole

- temperature difference with much greater fidelity than before (Huber and Caballero, 2011; Lunt et al., 2012; Kiehl and Shields, 2013). A remaining problem is to understand what combination of radiative forcing and climate sensitivity gave rise to these very elevated temperatures in the first place (Caballero and Huber, 2013). While early Cenozoic  $CO_2$  concentrations are understood to have been higher than modern (Royer, 2014), achieving a good match with reconstructed temperatures using  $CO_2$  alone as the warming agent requires extremely high model  $CO_2$  levels (Lunt et al., 2012), exceeding even the rather loose
- constraints imposed by the  $CO_2$  proxies.

Non-CO<sub>2</sub> greenhouse gases such as methane and nitrous oxide may have contributed some of the warming (Beerling et al., 2011), but we lack suitable proxies to constrain their concentrations. Another hypothesis is that reduced aerosol loading during the Eocene could play an important role, particularly via its effect on cloud properties (Kump and Pollard, 2008; Kiehl and Shields, 2013). In a pre-industrial climate aerosol inventories are largely controlled by biological productivity (Andreae,

- 2007). In a very warm climate, it is possible that temperature limits could be exceeded over large areas of the tropics and subtropics (Huber, 2008) that would sharply limit primary production there, reducing emissions of aerosol precursors to the atmosphere. Further, increased ocean stratification could put further stress on ocean biota, limiting the emission of sulphate precursors (Behrenfeld et al., 2006), while stomatal closure in an elevated CO<sub>2</sub> environment could reduce the emission of organic aerosol precursors from terrestrial plants (Acosta-Navarro et al., 2014). Reduced abundance of
- aerosols and cloud condensation nuclei is generally thought to lead to larger cloud droplets and reduced cloud albedo (Lohmann and Feichter, 2005). A global reduction in aerosol loading would thus lead to lower planetary albedo and a warmer climate. However, we again lack proxies to constrain paleo-aerosol abundances, and cloud-aerosol interactions themselves are still imperfectly understood (Stevens and Feingold, 2009), so this scenario remains highly speculative: given the current level of understanding, a substantial decrease in aerosol loading and cloud albedo during the Eocene can be neither
- **confirmed nor ruled out with any degree of confidence.** Nonetheless, the problem of disentangling the mix of radiative forcing agents that gave rise to Eocene warmth remains of crucial importance given its implications for climate sensitivity and thus predictions of future climate change (Caballero and Huber, 2013).

Here, we explore the potential for constraining the radiative forcing mix responsible for Eocene warmth *indirectly*—that is, without relying on direct proxies for aerosols or non-CO<sub>2</sub> greenhouse gases. Different combinations of forcing agents supporting the same global-mean surface temperature may leave different signatures in other climatological fields that are potentially detectable in the geological record. We focus specifically on the distinction between warming by greenhouse gases—which affect the longwave radiation—and warming by reduced cloud albedo, which affects solar radiation. We do this by comparing simulations of the Eocene climate warmed with respect to the present solely by increased CO<sub>2</sub>—which we refer to as the greenhouse gas (GHG) scenario—with simulations using modern pre-industrial CO<sub>2</sub> warmed solely by reduced cloud albedo, but with indistinguishable global-mean surface temperature. As in previous work (Kump and Pollard, 2008; Kiehl and Shields, 2013) we reduce cloud albedo by globally increasing the prescribed size of cloud droplets; we refer to these the "low CO<sub>2</sub>—thin clouds" (LCTC) scenario (because the clouds are optically thinner in the shortwave spectrum). We pay particular attention to differences in surface temperature patterns and to the hydrological cycle, which is known to respond differently

to warming by longwave and shortwave forcing (O'Gorman et al., 2012). A similar discussion arises in the context of geoengineering proposals to artificially increase Earth's planetary albedo in order to offset some of the warming due to anthropogenic
CO<sub>2</sub> emissions; it is well known that the resulting climate has a significantly different hydrological cycle compared with a climate subject to the same net radiative forcing but due to a more modest increase in CO<sub>2</sub> (Mcnutt et al., 2015).

Differences in surface temperature patterns and in hydrological regime could potentially be detectable in the terrestrial record, particularly in vegetation. However, it is not our goal here to provide an exhaustive comparison between our

simulations and the extant terrestrial proxy record. Given the broad-brush approach we take, employing a globally-

uniform increase in cloud droplet radius, it is unlikely that such a detailed comparison would be very meaningful in any case. Instead, we focus on quantifying the differences between the two climates and understanding the climate dynamical processes that lead to these differences, with the aim of assessing whether these differences are large and robust enough to warrant further, more detailed work in this direction.

5 Our modelling approach is further described in Section 2. Section 3 presents an overview of the climatology in the GHG simulation and its changes in the LCTC case. Section 4 discusses the global-mean change in precipitation and its relation to energetic constraints. Regional climate differences between the two simulations and their relation to circulation changes are discussed in Section 5. Finally, Section 6 summarizes our conclusions.

# 2 Methods

#### 10 2.1 Model and simulations

We employ the Community Atmospheric Model 3.1 (CAM3) developed by the National Center for Atmospheric **Research** (NCAR) (Collins et al., 2006b) at T42 resolution coupled to **the Community Land Model 2 (CLM2) and to a** slab ocean **with uniform 50 m depth**. Ocean heat transport **and seasonal heat storage are** approximated through a prescribed seasonally-varying energy convergence field ("q-flux") derived from a fully-coupled Eocene simulation run to equilibrium for a corre-

15 sponding climate (Huber and Caballero, 2011). The model is configured with Eocene geography and land surface cover as described in Sewall et al. (2000). Simulation climatologies are computed from 30 years of monthly output obtained after the model has reached equilibrium.

We focus on two end-member simulations of the Eocene: a **GHG** run warmed only by increasing  $CO_2$  and an extreme LCTC run warmed only by decreasing the planetary albedo. In the **GHG** simulation  $CO_2$  is set to 4480 ppm and the **effective** 

- **cloud droplet radius** takes its default value (8  $\mu$ m over land and 14  $\mu$ m over ocean). Various aspects of this simulation have been previously described in previous work (Caballero and Huber, 2010; Huber and Caballero, 2011; Caballero and Huber, 2013). In the LCTC simulation, CO<sub>2</sub> takes its pre-industrial value of 280 ppm while **effective cloud droplet radius** is increased by multiplying the default values by a uniform factor of 2.5, yielding 20  $\mu$ m over land and 35  $\mu$ m over ocean Note that this retains the difference between land and ocean values, differently from Kiehl and Shields (2013) who used a single
- 25 effective cloud droplet radius globally. There is little knowledge about the actual aerosol concentration during the Eocene, but present-day observations in remote regions indicate a land-sea difference in droplet size can be expected even in the absence of anthropogenic emissions (Bréon and Colzy, 2000).

These are very large and likely unrealistic droplet sizes, chosen simply because they lead to the same global-mean surface temperature as in the **GHG** run. **In addition, cloud microphysics aside from the effective cloud droplet radius is** 

30 unchanged, which unrealistically neglects cloud lifetime effects and imposes a further limitation in the interpretation of the results. Our intention is to compare two end-member states which maximise the difference between the two warming agents. Intermediate combinations of  $CO_2$  and cloud albedo (e.g. Kiehl and Shields, 2013) will presumably have intermediate climate impacts; to test this hypothesis, we also conducted a third simulation where  $CO_2$  is set to 1120 ppm and droplet radii are scaled by a factor of 1.6, and confirmed that the resulting climate differs from the **GHG** in a qualitatively similar way as described below for the extreme LCTC case, but with smaller overall amplitudes. In the interest of conciseness we focus here only on the end-member cases.

- Comparison of the annual-mean surface temperature and precipitation fields in the GHG case (which are presented in the following section) with the corresponding fully-coupled Eocene simulation of Huber and Caballero (2011) show good agreement, with zonal-mean temperature and precipitation differences on the order of ~2 K and ~1 mm day<sup>-1</sup> respectively. As discussed in Huber and Caballero (2011), the coupled model reproduces the available proxy temperature reconstructions reasonably well, with no appreciable bias in either the global-mean temperature or the mean equator-pole temperature difference. As shown in Carmichael et al. (2016) (where this simulation is referred to as
- CCSM\_H-16x), the precipitation field is also in broad agreement with a global compilation of precipitation proxies, albeit with some mismatches particularly in the Southern Hemisphere high latitudes. This provides confidence that the GHG simulation is a reasonable approximation to the Eocene climate at least in the annual mean. However, seasonal biases are larger, with a reduction of the annual temperature range in mid- to high latitudes of around 5 K in the GHG simulation compared to the full coupled model. This should be born in mind when interpreting the climatologies below.
- These seasonal biases could possibly be corrected by suitably adjusting the q-flux and/or slab depth. Nonetheless, since our main focus in this paper is on the differences between two simulations using the same slab-ocean configuration, we believe our results should be at least qualitatively robust to the details of the q-flux specification.

# 3 Climatologies

This section describes the main climatological features of the two simulations, to be further analysed and interpreted in subse-20 quent sections.

Zonal- and annual-mean profiles of surface temperature and precipitation for the two simulations are presented in Figure 1. Despite their very different radiative forcing agents, the two simulations have almost identical zonal-mean temperatures, **differing at most by around 0.9 K**. In the global mean, the LCTC simulation is cooler by 0.3 K than the **GHG** simulation. Precipitation, on the other hand, is significantly larger in the LCTC case, especially in the tropics and in the subtropical flanks of the major midlatitude precipitation zones.

As shown in Figure 2b, however, regional differences in surface temperature are large. Land areas are cooler in the LCTC run by up to 5 K, particularly in subtropical regions with little cloud cover (Figure 2e), **the annual-mean climatology in the LCTC case is shown in Figure S1 in the supplementary material**. Offsetting warming develops over the oceans, notably in twin horseshoe-shaped regions of the Pacific in both hemispheres. These temperature responses are accompanied by large changes

in the low-level circulation (Figure 2b). The LCTC case exhibits a cyclonic anomaly over the northern Pacific, with a similar but weaker response over the southern Pacific. The cyclonic anomaly acts to weaken the prevailing subtropical **anticyclones** seen in the **GHG** case (Figure 2a). This is reminiscent of the weakening of the subtropical anticyclones seen in the transition from summer to winter in the modern day, which is also accompanied by a cooling of the continents relative to the oceans. While global-mean temperature is **very similar** in the two simulations, global-mean precipitation is **3.8 mm day**<sup>-1</sup> **in the GHG simulation and 0.43 mm day**<sup>-1</sup> or about 11% higher in the LCTC case. This must be considered a substantial increase: Carmichael et al. (2016, their Figure 2c) show a robust linear scaling of precipitation with temperature of around 0.06 mm day<sup>-1</sup> K<sup>-1</sup>, implying that a precipitation increase of this magnitude would require a CO<sub>2</sub>-driven warming in excess

of 7 K (comparable to that across the Paleocene-Eocene Thermal Maximum (PETM) hyperthermal event, see Pagani et al., 2014). For comparison, a present-day simulation using CAM3 (Collins et al., 2006a) has global-mean temperature around 15 K cooler and precipitation around 0.9 mm day<sup>-1</sup> lower than the GHG simulation, in agreement with the scaling above.

As shown in Figure 2d the spatial distribution of the precipitation change is highly non-uniform, with regional increases
well in excess of 100%. The greatest precipitation increase is concentrated in the central and eastern equatorial Pacific, partly offset by a large decrease in the western Pacific Warm Pool region. Precipitation also increases across large swaths of the subtropical to midlatitude Pacific and Atlantic Oceans and adjacent land areas.

Cloud cover in the **GHG** simulation (Figure 2e) shows broad-scale structures similar to the modern day, with abundant cloud cover over the Pacific Warm Pool and intertropical convergence zones (ITCZ) and over the midlatitude oceanic storm

- tracks and extensive stratocumulus decks in the eastern subtropical margins of the main ocean basins. Though our LCTC simulation only prescribes changes in cloud droplet size and CO<sub>2</sub>, the resulting dynamical and thermodynamical changes lead to changes also in cloud fraction (Figure 2f). In particular, there is a substantial decrease in cloud fraction in the North Pacific subtropics—in the same region showing a large sea-surface temperature (SST) warming (Figure 2b)—and increased cloud cover over southwestern North America, associated with increased precipitation there (Figure 2d).
- Figures 2g,h examine the atmospheric jets and storm tracks in the two simulations. As in the modern climate, zonal winds in the Northern Hemisphere of the **GHG** simulation are concentrated into separate jet streams spanning the Pacific and Atlantic basins; differently from modern, the Pacific jet exhibits a marked meridional tilt. **The Pacific jet's modern structure is well** represented in modern-day simulations using CAM3 (Hurrell et al., 2006), so the jet's meridional tilt in our GHG simulation appears to be an intrinsic feature of this climate state, potentially related to its high CO<sub>2</sub> (Grise and Polvani,
- 2013) and to the structure of Asian orography, which lacks the modern Tibetan Plateau and instead presents a more continuous north-south obstacle with modest elevation (Brayshaw et al., 2009; Löfverström et al., 2016). The jets are associated with regions of enhanced eddy variability—storm tracks—which are grossly identified here by the sub-monthly eddy kinetic energy EKE =  $(u'^2 + v'^2)/2$ , where u and v are zonal and meridional components of the wind and primes denote sub-monthly deviations from monthly climatology. Lower-tropospheric jets and storm tracks are intimately connected via
- wave-mean flow interaction (Shaw et al., 2016). In the **GHG** run, the North Pacific storm track is further northward than observed in the modern climate (Shaw et al., 2016), consistent with the strong meridional tilt of the Pacific jet. In the LCTC case, however, the North Pacific jet and storm track show a marked equatorward shift, with enhanced winds and EKE in the subtropical basin and reductions further north. In the Southern Hemisphere the jet and storm tracks are more zonally continuous, but a similar equatorward shift can be observed in the LCTC case. Along the equator, the LCTC case shows a low-level westerly
- wind anomaly in the western Pacific, consistent with a weakening of the Walker Cell and an eastward shift of precipitation

towards the central Pacific as noted above. There is also a marked reduction in EKE in the western equatorial Pacific, in the same region where precipitation decreases. Much of the sub-seasonal variability in the tropics is due to the Madden-Julian Oscillation (MJO). As discussed elsewhere (Caballero and Huber, 2010; Arnold et al., 2014; Carlson and Caballero, 2016), warm climates show a strongly enhanced MJO in a range of climate models including the one used here. It is possible that the

5 MJO is more muted in the LCTC simulation, perhaps because of changes in the Walker Cell, but we do not investigate this issue further here.

Figure 3 compares the two simulations' annual and diurnal surface temperature ranges, which play an important role in determining both floral and faunal species ranges (Pearson and Dawson, 2003). Monthly-mean temperatures during the warmest month in the GHG simulation are above 320 K over large parts of the subtropical continents

- (Figure 3a), potentially exceeding the survivability limits of vegetation there (Huber, 2008). These regions are cooler in the LCTC case (Figure 3b) due to generally low cloud cover there—which limits the warming effect of thinner clouds—combined with the lower CO<sub>2</sub>. This LCTC cooling, though modest (generally around 3-5 K) could facilitate the survival of plants over the summer. Palynological evidence from northwestern South America (Jaramillo et al., 2006) indicates high floral biodiversity there, suggesting little environmental stress; however, this evidence comes from
- a near-equatorial region where both simulations show more moderate warm-month temperatures in addition to high precipitation. Floral reconstructions directly relevant to the extremely warm subtropical regions appear to be lacking at present. At higher latitudes, summer temperatures are somewhat warmer in the LCTC than the GHG case, presumably because high cloud cover there allows the warming effect of thinner clouds to dominate over the reduction in CO<sub>2</sub>.
- Cold-month temperatures are above freezing everywhere in the GHG simulation except for parts of eastern Asia and North America (identified by white contours in Figure 3c). In the LCTC simulation, cold-month temperatures are generally cooler, consistent with a weaker warming effect by thin clouds during winter when insolation is low. This leads to a moderate expansion of the regions experiencing sub-freezing winter temperatures, though importantly both the Arctic Ocean and Antarctica remain above freezing. There is substantial fossil evidence for subtropical, frostintolerant flora and fauna surviving in western continental interiors and along the margins of the Arctic Ocean during
- the early Eocene (Markwick, 1998; Greenwood and Wing, 1995), and both GHG and LCTC simulations are largely compatible with such evidence.

Lastly, we turn to the diurnal surface temperature range, estimated as the difference in monthly-mean maximum and minimum temperatures at each grid point (these quantities are not available in the model output over oceans, so only land values are shown in Figures 3e,f). The annual-mean diurnal temperature range in the GHG simulation

- (Figure 3e) is large over parts of the subtropical continents, particularly in western Asia and North America where it locally exceeds 16 K, consistent with low cloudiness in those regions. In the LCTC case, the diurnal temperature range is modestly larger over most continental regions; this enhancement in the LCTC case can be ascribed to greater daytime heating due to reduced cloud albedos and greater nighttime cooling due to reduced CO<sub>2</sub>. The enhancement is markedly stronger over subtropical and midlatitude Asia, where it is up to 50% greater than in the GHG case, which we ascribe
- to the substantial decrease in cloud cover in that region (Figure 2f). Conversely, subtropical North America experiences

a diurnal temperature range around 20% smaller than in the GHG case, consistent with increased cloud cover there which overcompensates for the reduced cloud albedo.

#### 4 Global-mean precipitation and energetic constraints

As noted above, global-mean precipitation in the LCTC case is around 11% higher than in the **GHG**. A precipitation increase 5 in response to a simultaneous drop in  $CO_2$  and planetary albedo is consistent with simulations of geoengineering scenarios, where increased  $CO_2$  and planetary albedo lead to lower precipitation (Bala et al., 2008). In this section we account for the global-mean precipitation change in our simulations from the perspective of energy budget constraints (O'Gorman et al., 2012).

The global-mean atmospheric energy budget can be written, assuming steady state, as

$$SW_{TOA} - SW_{srf} + LW_{TOA} - LW_{srf} - LH - SH = 0, (1)$$

where SW and LW refer to shortwave and longwave radiation with subscripts TOA and srf indicating top-of-atmosphere and surface fluxes respectively, while LH and SH are surface latent and sensible heat fluxes respectively. All fluxes are taken to be positive downwards. Given the atmospheric steady state assumption,  $LH = -L_vP$ , where P is global-mean precipitation and  $L_v$  the latent heat capacity. If the planet as a whole is also in steady state, then  $SW_{TOA} + LW_{TOA} = 0$  and (1) reduces to the surface energy budget

$$SW_{srf} + LW_{srf} + LH + SH = 0.$$
 (2)

For climates in planetary energy balance, like those studied here, the atmospheric and surface energy budgets are thus equivalent, and provide alternative and complementary perspectives on the mechanisms controlling changes in precipitation (Allen and Ingram, 2002; Pierrehumbert, 2002). We examine both perspectives here.

- The terms in (1) for the **GHG** case are shown in Figure 4a. **The** *LH* **is the largest component**, but shortwave absorption 20  $(SW_{TOA} - SW_{srf})$  also makes a large contribution as can be expected in this very warm and thus moist climate. Sensible heating makes a much smaller contribution. The heating is entirely balanced by net longwave cooling  $(LW_{TOA} - LW_{srf})$ . Changes when going to the LCTC case are shown in Figure 4b. Net longwave cooling increases by about 17 W m<sup>-2</sup> in the LCTC run. This increased cooling is a combination of clear-sky and cloud effects. In clear skies, lowering CO<sub>2</sub> while keeping temperature and humidity fixed yields stronger outgoing longwave radiation at the TOA (which increases atmospheric cooling)
- and weaker downward radiation at the surface (which decreases cooling). The former effect dominates the latter, however (Pendergrass and Hartmann, 2014), so the net result is increased cooling. The contribution of cloud effects can be estimated using the cloud radiative effect (CRE, difference between all-sky and clear-sky radiation). Net atmospheric longwave CRE the difference between TOA and surface CRE—is about 13 W m<sup>-2</sup> in the **GHG** case, implying clouds have a net heating effect in the longwave. This heating drops to 7 W m<sup>-2</sup> in the LCTC case, which means that **cloud changes**—including the
- 30 general reduction in cloud cover seen in Figure 2f as well as changes in cloud LW opacity due to increased effective drop radius—make a substantial contribution of around 6 W m<sup>-2</sup> to the 17 W m<sup>-2</sup> total increase in atmospheric cooling. Finally,

Figure 4b also shows that increased longwave cooling is balanced mostly by latent heating, with sensible heating and shortwave absorption playing minor roles. In summary, the atmospheric energy budget perspective indicates that precipitation increases in the LCTC run to compensate for increased longwave cooling due mostly to the clear-sky effect of reduced  $CO_2$ , with **a** substantial contribution also from changes in cloud fraction and emissivity.

- From the surface budget perspective (2), Figure 4b (noting the reversed sign convention) shows that increased shortwave heating of the surface in the LCTC case—due to the lower cloud albedo—is mainly compensated through increased latent heat flux and also increased longwave surface cooling (due to reduced downwelling radiation, as noted above). From the surface perspective, then, precipitation increases in the LCTC case mostly to balance increased surface solar heating. This points to a rather different physical picture than the atmospheric perspective, in which shortwave fluxes play a negligible role. Given the
- equivalence of (1) and (2), the two physical pictures must of course be consistent. A plausible hypothesis for how consistency is achieved runs as follows: atmospheric destabilization by radiative cooling due to reduced  $CO_2$  accelerates convection, increasing rainfall and also mixing drier air down into the boundary layer; this drying increases evaporative demand at the surface, and the extra energy required to maintain surface temperature against evaporative cooling is supplied by increased solar absorption due to reduced cloud albedo. Fully reconciling the two perspectives in a causal, mechanistic way would
- require considerably more work going beyond the scope of this paper. However, some evidence supporting this hypothesis is shown in the following section.

#### 5 Atmospheric circulation and regional climate response

While the energetic constraints discussed above help explain the global increase in precipitation in the LCTC case, they place no constraints on its spatial distribution, which is highly heterogeneous (Figure 2d). These regional changes in precipitation are
accompanied by pronounced changes in the surface temperature pattern (Figure 2b) and the atmospheric general circulation (Figure 2h). In this section we explore how these various changes are interrelated. To do this, it is useful to think of the transition from the GHG to the LCTC climate as if it occurred in 3 stages: first CO<sub>2</sub> is reduced instantaneously, producing a fast adjustment in the atmosphere and land before SST has time to change (Sherwood et al., 2015); then cloud albedo is instantaneously reduced, producing a further fast adjustment; and finally the SST slowly adjusts to its final equilibrium pattern.
Circulation changes linked to the fast adjustments will condition the evolution of the SST pattern, which in turn will affect the circulation.

We make this conceptual picture quantitative by running 3 fixed-SST experiments. Two employ prescribed (seasonallyvarying) SST from the **GHG** run; one uses **GHG** values of cloud drop radius while  $CO_2$  is reduced to 280 ppm while the other uses **GHG**  $CO_2$  and cloud drop radius increased by a factor of 2.5. The third run uses **GHG** values of  $CO_2$  and cloud drop

radius but prescribes the SST from the LCTC case. Comparing these fixed-SST simulations with the GHG run allows us to

30

separately quantify the effect of changing CO<sub>2</sub>, cloud albedo and SST pattern.

Figure 5 shows the surface temperature and low-level circulation responses in the three simulations. The sum of all changes (Figure 5d) gives a reasonable match to the full response in the LCTC run (Figure 2b), though with somewhat higher amplitude,

suggesting that this linear decomposition is an adequate approximation. The direct response to reduced  $CO_2$  (Figure 5a) involves strong cooling of the extratropical continents and a basin-scale cyclonic anomaly over both the North and South Pacific, with westerly anomalies spanning the tropics and lower midlatitudes and easterly anomalies further north. This is accompanied by an equatorward migration of the upper level jets in both hemispheres, consistently with previous work (Grise

5 and Polvani, 2013). Cooling of midlatitude land relative to the ocean accompanied by cyclonic anomalies over the ocean is reminiscent of the negative phase of the "cold ocean–warm land" pattern (Wallace et al., 1996). It is also consistent with the work of Molteni et al. (2011), who show cyclonic anomalies over the North Pacific in simulations with permanently increased land-ocean temperature contrast and explore alternative dynamical scenarios to account for them.

The direct response to cloud albedo reduction (Figure 5b) is a warming of the continents, particularly at high latitudes. This

- 10 warming is highly seasonal, peaking in the summer, unlike the  $CO_2$  response which is more even through the year. Somewhat surprisingly, the circulation response to continental warming is weak and not obviously anticyclonic except in the South Pacific. The reasons for this weak response are unclear; it is perhaps due to the seasonal and high-latitude nature of the warming, but we do not explore the issue further here.
- Finally, the response to changing SST pattern (Figure 5c) features a strong cyclonic anomaly over the North Pacific—
  where the SST anomaly is strongest—and a weaker cyclonic anomaly over the South Pacific. In the extratropics of both hemispheres, the circulation responses align with those induced by CO<sub>2</sub> alone, and similarly yield an equatorward shift of the lower-tropospheric jets. There is also a strong response in the tropics, in particular an westerly anomaly in the tropical west Pacific indicating a weakened Walker cell consistent with the substantial warm anomalies in the central Pacific.
- Precipitation changes from the GHG for the 3 runs are presented in Figure 6. The sum of all changes (Figure 6d) again captures the change seen in the full LCTC case (Figure 2d) reasonably well. Global-mean precipitation increases by 0.37 mm day<sup>-1</sup> in response to CO<sub>2</sub> alone, by 0.05 mm day<sup>-1</sup> in response to cloud albedo alone and decreases slightly in response to SST pattern. Thus, most of the 0.42 mm day<sup>-1</sup> increase seen in the LCTC run is due to the direct effect of CO<sub>2</sub>. Reduced cloud albedo drives little change in precipitation by itself—instead, as discussed at the end of Section 4, it serves to close the surface energy balance, supplying surface solar heating to offset increased evaporative cooling. While CO<sub>2</sub> drives most of the 25 global-mean precipitation change, its spatial pattern (Figure 6a) shows anomalies concentrated in the eastern Pacific and over the Indo-Pacific warm pool, which is very different from that in the final state (Figure 2d). Changes in SST pattern clearly play
- a major role in redistributing tropical precipitation into the central Pacific (Figure 6c), consistent with warm SST anomalies there enhancing low-level convergence as is evident in Figure 5c.
- Taken together, these results indicate a key role for the SST pattern in mediating the transition from the **GHG** to the LCTC 30 climate state. So what gives rise to the SST anomaly? Some insight into this question is provided by the work of Vimont et al. (2001), who argue that a cyclonic circulation anomaly in the extratropical North Pacific will yield an SST "footprint" which has precisely the horseshoe structure we find here (compare their Figure 1 with Figure 2b). This happens because the anomalous surface winds affect surface energy fluxes. This interpretation is supported by Figure 7, which shows the net surface energy flux change from the **GHG** to the fixed-SST low CO<sub>2</sub> simulation. The pattern of ocean heating and cooling induced by these
- 35 surface flux anomalies clearly aligns well with the SST anomaly (Figure 5c). Note in particular that the surface energy flux

anomaly will tend to warm the central Pacific relative to the rest of the equatorial zone, driving a shift in precipitation into the central Pacific.

A further important point highlighted by Figure 5c is that the SST anomaly itself produces an extratropical circulation response that superposes constructively on the pre-existing  $CO_2$ -induced cyclonic circulation anomaly. Given the generally

- weak atmospheric response to extratropical SST anomalies (Kushnir et al., 2002), it is most likely that this extratropical circulation response is driven from the tropics, in particular by warm SST anomaly in the central Pacific which—as noted above—promotes large precipitation anomalies there, much like in an El Niño event. Such tropical heating anomalies are known to robustly induce cyclonic circulation anomalies in the extratropical North Pacific (Alexander et al., 2002). Furthermore, it is well known from both observations (Caballero, 2007) and model studies (Tandon et al., 2013) that El Niño events and their
- associated tropical heating anomalies drive an equatorward shift of the extratropical jets and storm tracks, in agreement with what we find here (Figure 2h).

In summary, the picture that emerges is that the direct effect of reduced  $CO_2$  initially drives a basin-scale cyclonic circulation anomaly in each hemisphere of the Pacific Ocean; this circulation anomaly then drives SST anomalies which reinforce the initial response to  $CO_2$ . The direct response to cloud albedo appears to play a minor role in driving regional climate changes. This

picture points to an important limitation of our modelling approach, which uses a slab ocean model. With a dynamic ocean, the westerly wind anomalies along the equator (Figure 7) would likely drive a deepening of the ocean thermocline in the eastern equatorial Pacific, shifting the mean climate towards a more El Niño-like state. This response and its global consequences are an important target for future work using a fully-coupled modelling approach.

# 6 Conclusions

- We have studied the differences in circulation and hydrological cycle resulting from two extreme scenarios by which Eocene simulations can attain surface temperatures compatible with proxy reconstructions: one by warming exclusively by increased CO<sub>2</sub> (the **GHG** case), the other by warming exclusively via reduced cloud albedo (the LCTC case). The two simulations have essentially identical zonal-mean surface temperature, but the LCTC case has significantly higher precipitation. Analysis of the global-mean energy budget (Section 4) suggests that the increased precipitation can be viewed as resulting from greater radiative cooling of the atmosphere in the LCTC case due to its lower CO<sub>2</sub>. The spatial distribution of the precipitation increase is highly heterogeneous, and is concentrated largely in the central equatorial Pacific and in the lower midlatitudes. The midlatitude continents cool in the LCTC simulation, with compensating warming of the oceans particular in horseshoe-shaped patterns in both hemispheres of the Pacific. There are also major changes to the atmospheric circulation, with basin-scale cyclonic anomalies appearing in both hemispheres of the Pacific associated with equatorward shifts of the storm tracks, and a
- 30 strong weakening of the Walker Cell in the tropics.

More detailed analysis (Section 5) suggests that these various anomalies are dynamically interrelated. Lower  $CO_2$  in the LCTC case leads to continental cooling, which in turn generates cyclonic circulation anomalies over the Pacific. We propose that these cyclonic anomalies in turn leave a horseshoe shaped "footprint" on SSTs via their effect on surface turbulent fluxes

(Vimont et al., 2001). This footprint reaches into the central tropical Pacific and promotes increased convection there which, much as in a modern El Niño event, affects the extratropical circulation and enhances the pre-existing cyclonic anomalies. This self-reinforcing mechanism leads to the pronounced regional climate anomalies noted above, which may also affect adjacent land areas.

- Our work is intended as an initial exploration, and much work remains to be done to remove the limitations imposed by our choice of a relatively simplified modelling approach. A key limitation is our use of a slab ocean. As noted in Section 5, the ocean will dynamically respond to the westerly surface stress anomalies along the equator (Vimont et al., 2001), possibly leading to a climate state with a permanently reduced tropical thermocline tilt and a more El Niño-like climate. If this were the case, the ocean dynamical response would in fact further strengthen the tropical anomalies found here, which already resemble an El
- Niño-like response. Another important caveat is that circulation responses to changes in radiative forcing—and their associated regional climate changes—are sensitive to biases in the unperturbed state (Shepherd, 2014). For example, if the Pacific jet were more zonally oriented in reality than in our GHG Eocene simulation (Figure 2g), its response in the LCTC scenario could be quite different. A further limitation of our approach is the specification of a uniformly increased cloud drop radius, leaving other aspects of cloud microphysics untouched. This is highly unrealistic in several ways: there is no particular reason to
- expect that different cloud types would respond in the same way to reduced aerosol loading; moreover, larger cloud drops are expected to coalesce more readily and thus reduce cloud lifetimes, potentially leading to reductions in cloud cover which would be different for different cloud types. In addition, all the above effects would depend on the precise nature and composition of natural aerosol in the Eocene, which remains essentially unknown. Different hypotheses for aerosol composition and cloud microphysics could lead to very different spatial structures of the resulting radiative forcing, with potentially large impacts on
- the circulation and regional climates.

Despite these important caveats, we conclude that differences in the radiative forcing agent driving Eocene warmth could at least in principle lead to large differences in regional climates leaving potentially detectable traces in the geological record. We have explored here two end-member scenarios, one with very high  $CO_2$  (higher than suggested by current  $CO_2$  proxy reconstructions) and another with pre-industrial  $CO_2$  (which is almost certainly lower than in the Eocene). A more realistic

- scenario would involve some intermediate mix of warming by  $CO_2$  and by cloud effects (Kiehl and Shields, 2013), which would be expected to yield smaller climate differences than those found here. However, given the large uncertainties discussed above, it remains possible that even an intermediate mix of warming agents could lead to responses as large or larger than those in our study. Future work with a range of models including more complete representations of ocean dynamics and cloud-aerosol interactions is required to settle this question.
- Acknowledgements. We thank Qiang Fu, Jack Scheff and Johan Nilsson for useful discussion and comments. The Swedish National Infrastructure for Computing (SNIC) at the National Supercomputing Centre (NSC), Linköping University, provided the high performance computing resources to perform the simulations.

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

Figure 1. Annual- and zonal-mean surface temperature (a) and precipitation (b) in the GHG case (magenta) and LCTC case (cyan).