# Peer review of "Atmospheric circulation and hydroclimate impacts of alternative warming scenarios for the Eocene"

_Climate of the Past, 2017_

## Referee Comment (RC1) · Anonymous Referee #1 · 29 Mar 2017

General comments

Carlson and Caballero address an important and relevant question: Could changes in cloud properties have contributed to Eocene warming, in ways that might be reflected in the proxy record? The paper focuses on analyzing a new simulation where changes in cloud radiative properties are altered from a more conventional Eocene forcing scenario, and a smaller greenhouse gas forcing is needed to achieve the same change in global-mean temperature change.

The paper makes an interesting contribution to the literature. The question and analysis are well within the scope of Climate of the Past. The abstract provides a concise and complete summary. The authors thoroughly reference existing literature and appropriately highlight their contribution. The presentation is well-structured and clear, as is the language, and the wording is sufficiently precise.

The paper shows that the greenhouse-gas and cloud driven mechanisms of Eocene warming have similar temperature changes (both of which are consistent with the proxy record), but differ in their hydroclimate and circulation changes. The implication is that proxies of circulation and hydroclimate would be able to differentiate between these two forcing mechanisms.

The case for how hydroclimate and circulation proxies could differentiate the two warming scenarios is not completely clear-cut. In addition to the caveats already addressed in the concluding section, another important caveat is that this study is based on only one model. Different climate models have different biases, which are often larger for hydroclimate and circulation than they are for temperature. Also, the changes in cloud effective radius employed here are probably too large to be realistic (according to the discussion of Kiehl and Shields 2013). Another concerning feature of the simulation is that the radiative properties of clouds are altered but the microphysical properties are not, so in a sense the simulation is not completely self-consistent. This is particularly important because the conclusions are largely focused on hydroclimate, which might be affected by changes in microphysical properties of clouds. This lack of change in cloud microphysics is not described in the methods section (where it should be mentioned), it is discussed extensively in the conclusions.

In light of the implications for differentiating forcing mechanisms from the proxy record, a more extensive discussion of potential proxy records for regional climate and hydroclimate would be warranted (the beginnings of a discussion is included at the end of section 6). Is there existing relevant proxy evidence? Are there existing methods with which new records might be identified in new locations, or would new reconstruction methods need to be developed? Is it likely that useful records could be found over land, and if so in which regions? Or would useful records need to be identified in the ocean?

The title of the paper could be improved in two very minor ways. The first is that the analysis focuses on just one alternative mechanism for Eocene warming, but the title implies there are multiple. Second, hydroclimate seems to be a bigger focus than circulation, and is also is a more likely candidate for the focus of new proxy evidence than circulation. The order of the terms might be switched to reflect this.

The quality of the figures is generally adequate. Given the focus, it would be useful to see absolute fields for things like precipitation and aridity in the LCTC simulation, rather than just their differences from the Control case. This could be accomplished by showing the LCTC absolute fields rather than the control fields, but it would also be useful to have both in addition to the differences. Some of these could be included as a supplement. If one figure were to be omitted, I would suggest Figure 6. Its discussion is already limited in the text. In Figure 7, the caption states that fields are shown over land only, but panels a and c seem to have some faint signal over the ocean; it would be worth removing this, otherwise it gives the impression that changes over ocean are included and are uniform but small.

Specific comments

Page 3 Line 3: A more descriptive name than "control" might be devised for the greenhouse-gas driven Eocene scenario. More descriptive names for both scenarios might be "GHG" and "Cloud."

Line 4-5: Rather than cloud droplet radius, the relevant variables is the effective cloud droplet radius. It represents a weighted average over the distribution of cloud droplets.

Page 4 Line 5-8: Just how similar is the temperature for the two simulations? You might quantify this as the maximum difference in zonal-mean temperature. Figure 2b makes it seem like they could be non-trivial.

Page 5 Line 10-15: The jet streams vary among climate models and between models and observations in the present-day climate, and these variations are related to

changes in the jet with climate (see Barnes and Polvani 2013 for the differences across models). The appropriate comparison with present-day jet streams would be to CAM3 with modern boundary conditions and forcings, whereas Shaw et al (2016) focus on reanalysis data.

Line 21: Caution should be exercised in interpreting the MJO in coarse resolution climate models. There are many aspects of it that they are not able to represent with particularly good fidelity (e.g. Hung et al 2013).

Page 6 Line 19-23: It seems to me that 6 out of 17 Wmˆ-2 contributed by changes in cloud radiative effect is an important fraction of the total change in atmospheric radiative cooling.

Line 23: Rather than "cloud abundance," the cloud radiative effect might change due to the changes in cloud properties that are specified (effective cloud droplet radius).

Page 8 Line 6-7: Why is Fig 5d noisier than Fig 2d?

Minor comments Page 4 Lines 5,12: Figures 1 and 2 are shown in a different order from their reference in the text.

Line 22: "cyclones" should be "anticyclones"

Line 25: "essentially identical" is difficult to quantify, "very similar" might be more defensible

Page 5 Line 5: "cloud abundance" should be "cloud fraction"

Page 12 Line 24-26: The reference information for Carmichael et al (2015) seems to be out of date. The final published version of the paper came out in 2016 and has a different title.

Figure 3: I expected each color to balance across each panel, but they do not. The figure presentation might be a little bit clearer.

Additional references

Barnes, E. A., & Polvani, L. (2013). Response of the midlatitude jets, and of their variability, to increased greenhouse gases in the CMIP5 models. Journal of Climate, 26(18), 7117-7135.

Hung, M. P., Lin, J. L., Wang, W., Kim, D., Shinoda, T., & Weaver, S. J. (2013). MJO and convectively coupled equatorial waves simulated by CMIP5 climate models. Journal of Climate, 26(17), 6185-6214.

---

## Author Comment (AC1) · 30 Mar 2017

Thank you for the review and the useful suggestions, we believe we can address all the issues.

---

## Referee Comment (RC2) · Anonymous Referee #2 · 3 Apr 2017

This study uses a global climate model to try to answer the very interesting question of how warm worlds due to CO2 are different from warm worlds due to shortwave forcing (represented here by reduced cloud albedo), and how we might be able to see the difference in the geologic record. It is a fascinating and wide-ranging paper with a lot of insight.

However, on one key point (how to leverage the paleovegetation record) I think it is really missing the forest for the trees, so to speak. Namely, it focuses on regional water-induced vegetation differences between the two scenarios using a metric that the authors admit may not have much relevance. Yet direct CO2-induced vegetation differences would probably be much more one-sided, larger and easier to detect (and

by my reading of the literature would be likely to overwhelm the water-induced differences, calling their highlighting here into question.) Thus I recommend major revisions before full publication. I also have quite a few more minor science suggestions and writing suggestions under Minor afterward.

Major issue:

p2 li22, p9 li7-10, etc: The vegetation patterns might indeed carry some signature of the different hydrological cycle or regime, but an even stronger vegetation signature of LCTC vs. high-CO2 would be the direct CO2 effect on the vegetation itself! One would expect a very-high-CO2 world to be much greener than today's, with plants surviving in hydroclimate regimes that today cannot support them (because of the much lower transpiration losses in order to fix a given amount of CO2, i.e. the increased water-use efficiency.) In contrast, an LCTC world would not be expected to be particularly greener or browner than today's, except where induced by regional hydroclimate change. So the global vegetation extent and pattern could be a key *direct* indicator of the mix of forcings, independent of the forcings' effect on the water cycle. Certainly the vegetation pattern you prescribe (from the figure in Sewall et al 2000) is much greener than today's, with no unvegetated or desert-vegetation areas and very extensive forests, so you are almost implicitly assuming a major role for CO2 (or at least Sewall did.)

Similarly at end of page 3 and beginning of page 4 (and again p9 li12-13), yes the high CO2 might change the stomatal conductance and thus the terrestrial hydrology, but even more importantly it would dramatically change the vegetation itself (which is the thing you want to observe in the geologic record as you say on p9 li8-9), even before you get to the hydrology (!) You allude to this at p9 li13-14 when you cite the Roderick paper, but the way you put it is quite an understatement. Far from just "imperfect", the P/PET seems to have essentially nothing to do with the vegetation response when CO2 changes are involved... the CO2 utterly dominates. At least in models. See the plot in the Scheff manuscript, it is pretty stark (and largely backed up by the paleo vegetation data as they discuss.)

So I would strongly suggest getting rid of the whole PET-based analysis and instead looking at the land model's direct photosynthesis and/or leaf-area-index output if you want something you can qualitatively compare to paleovegetation records. Again, the contrast between the high-CO2 (fertilized) and the LCTC (unfertilized) case should be dramatic to say the least, unless the land model you are using is so old that it doesn't include CO2 effects. If you do this, you should add "and global vegetation" after "regional climates" at p11 li14.

I see that you are aware of this to some degree, and trying to get at this with the caveats and narrowing of the scope at p9 li21-24 and p10 li3-7 & li26-29. But it would seem cleaner and more of a clear demonstration to just plot the simulated changes in vegetation quantities like photosynthesis or LAI, instead of qualitatively fudging together the P/PET index changes with a vague expectation of additional greening/browning. It would also better highlight the most useful way of distinguishing high-CO2 from LCTC in the paleovegetation record - via the *direct* CO2 effects which have potentially global scope, rather than via the water effects which are much more regional and iffy as you point out.

Note, you can still keep all the precipitation and circulation stuff, since precipitation will still be relevant for understanding changes in P-E which will manifest themselves in proxies like paleo rivers and lakes, signatures of soil infiltration rates, etc. Precipitation also directly drives some of the paleomagnetic proxies. So I'm just talking about replacing sections 6 and 2.2 here, not touching the bulk of the paper which is much more relevant.

If you are really after differences in "eco-hydrological regime" (p9 li15) rather than vegetation per se, then I'd strongly recommend using P/Rnet (as suggested by the Milly and Dunne paper) rather than P/PET, to be conservative. But I'm not sure this is the best approach either, since there is no paleoproxy for eco-hydrological "regime" whatever that is. Instead there are proxies for specific tangible things like vegetation, rivers, lakes, infiltration rates, water isotopes, paleomag, etc. that all respond quite differently

to global climate change. In particular, vegetation responds very differently from P/Rnet or P/PET when CO2 is involved (as far as we can tell.)

Another way to see this is that satellite data shows the Earth is greening in recent decades (Zhu et al. 2016 Nature Climate Change) even while a P/PET calculation would tell you it has been drying (some of Fu's papers which analyzed historical data). This implies P/PET is not just "imperfect" for vegetation change under CO2/climate change, but grossly misleading.

Minor: (Note, some of these are on the PET parts, so they will not be relevant if you decide to replace the PET analysis as suggested above.)

p1 li10: Do you mean 11% greater absolute precip, or 11% larger precip *response* to warming? This should be clarified. See p4 li26-29 comment below.

pi li24-25: Can you at least include a few words about why one might think the Eocene would have fewer aerosols than the present? I.e., does this hypothesis just come out of nowhere or is there some qualitative speculation/cartoon behind it? Given how vegetated and warm the Eocene was, I would a priori think it would have more aerosols than present, via much-increased biogenic VOC emissions as well as more fires (more fuel.) Though, I suppose it would also have less dust due to the increased vegetation cover. So it could go either way.

p2 li30-31: The atmosphere and ocean models are specified here, but not the land model. What land model was used?

p2 li32: By prescribing a seasonally-varying q-flux but *also* using a slab mixed-layer ocean, you may be double-counting the seasonal storage and release of heat in the mixed layer... be careful here. What is the depth of the slab ocean? If you are going to use a seasonally-varying q-flux, your slab ocean should probably have minimal depth, since the seasonal cycle of the q-flux is *already* dominated by the fluxes in and out of the ocean every year due to seasonal warming and cooling of the mixed layer. (Does

this make sense?) In this case, you should also insert "and seasonal storage" after "transport" on line 31.

p3 li25: For the reader unfamiliar with hydrology and PET, you should also mention that capital Delta is the slope des/dT of the es curve (instead of just giving its numerical formula out of context.)

p3 li29-32: Scheff and Frierson (2015, the follow up to the 2014 paper) checked the monthly assumption more closely. They found that it's not bad at all for changes in PET (probably because the day-night warming difference isn't actually that big in most places) though it can throw off the absolute PET values by double-digit percentages. However, they only tested greenhouse-driven warming, so I don't know if the result would be valid for LCTC/shortwave warming. Presumably the LCTC world warms much more during the day than at night, so this time-resolution issue could actually become important.

In fact, the diurnal cycle of temperature could be another major constraint detectable in the paleoecological record - plant species composition may respond quite differently to warming with much-increased diurnal cycle (LCTC) than warming with unchanged or somewhat reduced diurnal cycle (CO2) because different species have different tolerances to min and max temperature. Again this would be independent of any water-cycle-driven vegetation change. Thus, it would be interesting for you to quantify the difference in diurnal Tmax and/or diurnal Tmin between your two Eocene end-members, not just the difference in Tmean. Either of these may be quite dramatic, even though the Tmean doesn't differ much. All of this may additionally hold for seasonal cycle (winter and summer temperatures rather than annual-mean temperature.) We have a good idea of winter temperatures from some of the proxies like crocodiles and palm trees surviving in the Arctic. Some SST proxies are also particularly sensitive to summer temperature so they could be useful here.

General: The first mention of Figure 1 right now (p4 li12) comes after the first mention of Figure 2 (p4 li5). So these two figures should be reversed and re-numbered accordingly. I.e. you need to re-number the current Figure 2 as Figure 1, and vice versa.

p4 li26-29: Not only is it a substantial increase due to this reasoning, it's also a substantial increase because it's on the same order or larger than what you would get by differencing LCTC with a Holocene run (or differencing high-CO2 with a Holocene run.) In general it might be more useful to contrast the differences of high-CO2 and LCTC with a common Holocene-like (i.e. low-CO2, normal-cloud) control, instead of contrasting the absolute precip values. I.e. by what factor or percentage is the LCTC-Holocene difference larger than the highCO2-Holocene difference. It will be a lot more than 11%. And it's that difference from present that we tend to notice when we think about Earth system change over time, not the absolute value.

p4 li29-34: This needs to cite Fig 2d.

p5 li3-4: No it doesn't "only" prescribe change in cloud droplet size - it also prescribes a big change in CO2 (of course.) The effects of the CO2 difference could also be important for the big cloud cover changes you see. Again, differencing both simulations with a common low-CO2 normal-cloud control would help in disentangling this.

p5 li10: Not to sound like a broken record here, but does a "modern" (Holocene-like) control simulation with this model also have this Pacific meridional tilt? You may just be seeing a general bias of the model, not a particular effect of the Eocene climate.

Exact same issue with "than observed in the modern climate" at li14-15, too. If Eocene model is different from modern observation, it's hard to tell whether that difference is due to "Eocene vs. modern" or whether it's due to "model vs. observation" (without further information or citations.)

p6 li1: Strictly speaking this equation will give a negative value for P, since LH as you just defined it will be a negative number (LH is upward, not downward) while Lv is

positive. So you may want a minus sign in this equation to be technically correct.

p6 li8 and Fig 3: It's not immediately clear what bars in Figure 3 to look at to see "Atmospheric heating". It seems for SW and LW it's the black (TOA-surface) bar, but for LH and SH it's actually the orange (surface) bar. So this would be easier to read if you made black bars for LH and SH as well, and explicitly pointed out in the text or the figure caption that the black (TOA-surface) is the (net) atmospheric heating in all cases. In this case you would also flip all of the orange bars to opposite sign, to be consistent with your definition of negative up, positive down (see previous comment.) Then, the coloring would be consistent across all four heat transfer mechanisms.

Alternatively you could keep your opposite-sign convention for the orange bars, but then you should rename the black bars to "atmospheric heating" rather than TOA minus surface, since strictly speaking they are TOA plus surface in the current graphical setup.

Also p6 li8: it's hardly "dominated" by the LH component, the SW component is actually almost as big! Rather you could just say that LH is the largest component (a lot more defensible.) "Dominated" means much larger than the others, not just larger.

p8 li4: I assume this is supposed to be "westerly anomaly" (i.e. east-pointing.)

p9 li17-18: This should have a citation (e.g. to Middleton and Thomas) so the reader can know who is doing the classifying.

p9 li19-20: Low relative humidity is also very important for this, I think (you could check this.)

p10 li27: As alluded in the major comment, "eco-hydrological differences" is too vague because plants, rivers, soils, precip, etc. all respond to CO2-driven climate change in such a different manner from one another. If you mean vegetation, just say vegetation. (If you mean something else, say something else.)

Figures 7a and 2c: It's hard to see the precipitation patterns over most of the planet, because most of the map is in the very light colors which don't distinguish from each

other easily. You may want to have your color scales saturate to dark at lower precipitation values (e.g. 9 instead of 18) to make the patterns easier to see. More than 9 mm/day is very wet by anyone's measure.

Figures 7a and 7c: Color values still seem to be plotted over the ocean, even though you are not trying to plot ocean points. This is a little confusing, and for 7a it worsens the above problem. Make sure the oceans are actually white here (or some other neutral color.)

Typos:

p2 li30: "research" needs to be capitalized in the name of NCAR.

p4 li22: should be "subtropical anticyclones" rather than "subtropical cyclones" (I presume?)

p5 li9: "zonal winds are in the Northern Hemisphere... are concentrated" should be "zonal winds in the Northern Hemisphere... are concentrated"

p8 li14: extra word "may" near the beginning of this line

---

## Author Comment (AC2) · 4 Apr 2017

Thank you for the review and for being very thorough and constructive, we believe we can address the issues.

---

## Author Response (AR1)

Response to reviewer #1:

We thank the reviewer for very helpful comments that made it possible for us to improve the manuscript. Point-by-point responses to the reviewer's comments are given below; the original comments are included in italics. All changes in the revised manuscript is marked in bold.

*General comments*
*Carlson and Caballero address an important and relevant question: Could changes in cloud properties have contributed to Eocene warming, in ways that might be reflected in the proxy record? The paper focuses on analyzing a new simulation where changes in cloud radiative properties are altered from a more conventional Eocene forcing scenario, and a smaller greenhouse gas forcing is needed to achieve the same change in global-mean temperature change.*

*The paper makes an interesting contribution to the literature. The question and analysis are well within the scope of Climate of the Past. The abstract provides a concise and complete summary. The authors thoroughly reference existing literature and appropriately highlight their contribution. The presentation is well-structured and clear, as is the language, and the wording is sufficiently precise.*

*The paper shows that the greenhouse-gas and cloud driven mechanisms of Eocene warming have similar temperature changes (both of which are consistent with the proxy record), but differ in their hydroclimate and circulation changes. The implication is that proxies of circulation and hydroclimate would be able to differentiate between these two forcing mechanisms. The case for how hydroclimate and circulation proxies could differentiate the two warming scenarios is not completely clear-cut. In addition to the caveats already addressed in the concluding section, another important caveat is that this study is based on only one model. Different climate models have different biases, which are often larger for hydroclimate and circulation than they are for temperature. Also, the changes in cloud effective radius employed here are probably too large to be realistic (according to the discussion of Kiehl and Shields 2013). Another concerning feature of the simulation is that the radiative properties of clouds are altered but the microphysical properties are not, so in a sense the simulation is not completely self-consistent. This is particularly important because the conclusions are largely focused on hydroclimate, which might be affected by changes in microphysical properties of clouds. This lack of change in cloud microphysics is not described in the methods section (where it should be mentioned), it is discussed extensively in the conclusions.*

Ans: This is an important caveat and it should be clear in the description of the simulations. We have now mentioned it in the methods section in the revised manuscript.

*In light of the implications for differentiating forcing mechanisms from the proxy record, a more extensive discussion of potential proxy records for regional climate and hydroclimate would be warranted (the beginnings of a discussion is included at the end of section 6). Is there existing relevant proxy evidence? Are there existing methods with which new records might be identified in new locations, or would new reconstruction methods need to be developed? Is it likely that useful records could be*

*found over land, and if so in which regions? Or would useful records need to be identified in the ocean?*

Ans: While we agree that a more comprehensive discussion of the extant and potential new proxies would be of great interest, we prefer not to delve too deeply in that direction here as it lies beyond our area of core expertise, which is in physical climate dynamics. Our goal here is simply to make a *prima facie* case from the climate dynamics perspective that the approach taken here could be a fruitful line of future, more detailed work, involving also the proxy community. We have added some lines in the Introduction to better clarify the aim and scope of our work.

*The title of the paper could be improved in two very minor ways. The first is that the analysis focuses on just one alternative mechanism for Eocene warming, but the title implies there are multiple. Second, hydroclimate seems to be a bigger focus than circulation, and is also is a more likely candidate for the focus of new proxy evidence than circulation. The order of the terms might be switched to reflect this.*

Ans: In response to comments from Reviewer 2, who convincingly argued for the potentially misleading nature of our approach to terrestrial hydroclimate, we have opted to remove that section (Section 6 in the original manuscript) from the paper. We feel that the climate dynamics-oriented material in Sections 4 and 5 is sufficiently extensive and novel to justify publication. We have also added a new figure and a substantial amount of new text to Section 3 dealing with seasonal and diurnal temperature ranges, which will be of interest to the proxy community. Given this refocusing, we feel that the paper's title adequately describes its contents.

*The quality of the figures is generally adequate. Given the focus, it would be useful to see absolute fields for things like precipitation and aridity in the LCTC simulation, rather than just their differences from the Control case. This could be accomplished by showing the LCTC absolute fields rather than the control fields, but it would also be useful to have both in addition to the differences. Some of these could be included as a supplement. If one figure were to be omitted, I would suggest Figure 6. Its discussion is already limited in the text. In Figure 7, the caption states that fields are shown over land only, but panels a and c seem to have some faint signal over the ocean; it would be worth removing this, otherwise it gives the impression that changes over ocean are included and are uniform but small.*

Ans: Thank you for pointing this out. We have included the absolute fields for the LCTC scenario in the supplementary material. The Figure 7 comment is no longer relevant since it has been excluded in the revised manuscript, see response to referee 2.

*Specific comments*
*Page 3 Line 3: A more descriptive name than "control" might be devised for the greenhouse-gas driven Eocene scenario. More descriptive names for both scenarios might be "GHG" and "Cloud."*

Ans: This is a good suggestion. We have changed the name of the control run to GHG but kept the name LCTC for the scenario with increased cloud droplet radius.

*Line 4-5: Rather than cloud droplet radius, the relevant variables is the effective cloud droplet radius. It represents a weighted average over the distribution of cloud droplets.*

Ans: This is an important distinction and it is corrected in the revised manuscript.

*Page 4 Line 5-8: Just how similar is the temperature for the two simulations? You might quantify this as the maximum difference in zonal-mean temperature. Figure 2b makes it seem like they could be non-trivial.*

Ans: The maximum difference between the GHG and LCTC zonal-mean temperatures is 0.93 K, we have added this to the text referring to Fig. 1. (Please note that lines 5-8 in page 4 of the original manuscript referred to a comparison between our GHG simulation and previously published (Huber and Caballero 2011) simulations with a coupled climate model under the same conditions; however, given the reference to Fig. 2b, we believe the reviewer is referring to the difference between the GHG and LCTC runs).

*Page 5 Line 10-15: The jet streams vary among climate models and between models and observations in the present-day climate, and these variations are related to changes in the jet with climate (see Barnes and Polvani 2013 for the differences across models). The appropriate comparison with present-day jet streams would be to CAM3 with modern boundary conditions and forcings, whereas Shaw et al (2016) focus on reanalysis data.*

Ans: This is a good point and we have added a reference that specifically looks at the jet streams in modern-day CAM3 (Hurrell et al. 2006), which shows that the modern zonal orientation of the Pacific jet is well represented in CAM3. We have elected to keep the reference of Shaw et al 2016 as well.

*Line 21: Caution should be exercised in interpreting the MJO in coarse resolution climate models. There are many aspects of it that they are not able to represent with particularly good fidelity (e.g. Hung et al 2013).*

Ans: This is a good point and in fact for a cold climate the model used here, CAM3, has very weak MJO-like variability. However, as is discussed in Carlson and Caballero 2016 in a warmer climate the tropical subseasonal variability is increasingly MJO-like. Although Carlson and Caballero 2016 looked at an aquaplanet ,a similar increase is seen in an Eocene configuration as was shown in Caballero and Huber 2012.

*Page 6 Line 19-23: It seems to me that 6 out of 17 Wmˆ-2 contributed by changes in cloud radiative effect is an important fraction of the total change in atmospheric radiative cooling.*

Ans: We agree and have qualified the statement with "makes a substantial contribution".

*Line 23: Rather than "cloud abundance," the cloud radiative effect might change due to the changes in cloud properties that are specified (effective cloud droplet radius).*

Ans: Yes, it is true that changes in LW cloud emissivity due to changed droplet size could also contribute to changes in cooling, and we have added this to the text.

*Page 8 Line 6-7: Why is Fig 5d noisier than Fig 2d?*

Ans: Through using the tool of a fixed SST model we have attempted to isolate the effect of the three different mechanisms. However, it is expected that the results are slightly different when comparing the full run (Figure 2d) and the linear sum of the three fixed SST runs (Figure 5d). The fact that the results are so similar is a motivation that the change we see is mainly due to these three effects, this conclusion is qualitative and based on the large scale similarity of Figure 2d and Figure 5d. The fact that the precipitation in the individual runs is not as noisy as the sum could suggest that the noise is an artifact of simply taking a linear sum.

*Minor comments*

*Page 4 Lines 5,12: Figures 1 and 2 are shown in a different order*
*from their reference in the text.*
*Line 22: "cyclones" should be "anticyclones"*
*Line 25: "essentially identical" is difficult to quantify, "very similar" might be more defensible*
*Page 5 Line 5: "cloud abundance" should be "cloud fraction"*
*Page 12 Line 24-26: The reference information for Carmichael et al (2015) seems to be out of date. The final published version of the paper came out in 2016 and has a different title.*

Ans: All of the minor comments above are very useful and have been addressed in the revised manuscript according to the suggestions above.

*Figure 3: I expected each color to balance across each panel, but they do not. The figure presentation might be a little bit clearer.*

Ans: This is a good point and it needed clarification. We have changed Figure 3 in order to make it clearer.

Response to reviewer #2:

We thank the reviewer for very helpful comments that made it possible for us to improve the manuscript. Point-by-point responses to the reviewer's comments are given below; the original comments are included in italics. All changes in the revised manuscript is marked in bold.

*This study uses a global climate model to try to answer the very interesting question of how warm worlds due to CO2 are different from warm worlds due to shortwave forcing (represented here by reduced cloud albedo), and how we might be able to see the difference in the geologic record. It is a fascinating and wide-ranging paper with a lot of insight.*

*However, on one key point (how to leverage the paleovegetation record) I think it is really missing the forest for the trees, so to speak. Namely, it focuses on regional water-induced vegetation differences between the two scenarios using a metric that the authors admit may not have much relevance. Yet direct CO2-induced vegetation differences would probably be much more one-sided, larger and easier to detect (and by my reading of the literature would be likely to overwhelm the water-induced differences, calling their highlighting here into question.) Thus I recommend major revisions before full publication. I also have quite a few more minor science suggestions and writing suggestions under Minor afterward.*

*Major issue:*
*p2 li22, p9 li7-10, etc: The vegetation patterns might indeed carry some signature of the different hydrological cycle or regime, but an even stronger vegetation signature of LCTC vs. high-CO2 would be the direct CO2 effect on the vegetation itself! One would expect a very-high-CO2 world to be much greener than today's, with plants surviving in hydroclimate regimes that today cannot support them (because of the much lower transpiration losses in order to fix a given amount of CO2, i.e. the increased water-use efficiency.) In contrast, an LCTC world would not be expected to be particularly greener or browner than today's, except where induced by regional hydroclimate change. So the global vegetation extent and pattern could be a key \*direct\* indicator of the mix of forcings, independent of the forcings' effect on the water cycle. Certainly the vegetation pattern you prescribe (from the figure in Sewall et al 2000) is much greener than today's, with no unvegetated or desert-vegetation areas and very extensive forests, so you are almost implicitly assuming a major role for CO2 (or at least Sewall did.)*

*Similarly at end of page 3 and beginning of page 4 (and again p9 li12-13), yes the high CO2 might change the stomatal conductance and thus the terrestrial hydrology, but even more importantly it would dramatically change the vegetation itself (which is the thing you want to observe in the geologic record as you say on p9 li8-9), even before you get to the hydrology (!) You allude to this at p9 li13-14 when you cite the Roderick paper, but the way you put it is quite an understatement. Far from just "imperfect", the P/PET seems to have essentially nothing to do with the vegetation response when CO2 changes are involved: : : the CO2 utterly dominates. At least in models. See the plot in the Scheff manuscript, it is pretty stark (and largely backed up by the paleo vegetation data as they discuss.)*

*So I would strongly suggest getting rid of the whole PET-based analysis and instead looking at the land model's direct photosynthesis and/or leaf-area-index output if you want something you can qualitatively compare to paleovegetation records. Again, the contrast between the high-CO2 (fertilized) and the LCTC (unfertilized) case should be dramatic to say the least, unless the land model you are using is so old that it doesn't include CO2 effects. If you do this, you should add "and global vegetation" after "regional climates" at p11 li14.*

*I see that you are aware of this to some degree, and trying to get at this with the caveats and narrowing of the scope at p9 li21-24 and p10 li3-7 & li26-29. But it would seem cleaner and more of a clear demonstration to just plot the simulated changes in vegetation quantities like photosynthesis or LAI, instead of qualitatively fudging together the P/PET index changes with a vague expectation of additional greening/browning. It would also better highlight the most useful way of distinguishing high-CO2 from LCTC in the paleovegetation record - via the \*direct\* CO2 effects which have potentially global scope, rather than via the water effects which are much more regional and iffy as you point out.*

*Note, you can still keep all the precipitation and circulation stuff, since precipitation will still be relevant for understanding changes in P-E which will manifest themselves inproxies like paleo rivers and lakes, signatures of soil infiltration rates, etc. Precipitation also directly drives some of the paleomagnetic proxies. So I'm just talking about replacing sections 6 and 2.2 here, not touching the bulk of the paper which is much more relevant.*

*If you are really after differences in "eco-hydrological regime" (p9 li15) rather than vegetation per se, then I'd strongly recommend using P/Rnet (as suggested by the Milly and Dunne paper) rather than P/PET, to be conservative. But I'm not sure this is the best approach either, since there is no paleoproxy for eco-hydrological "regime" whatever that is. Instead there are proxies for specific tangible things like vegetation, rivers, lakes, infiltration rates, water isotopes, paleomag, etc. that all respond quite differently to global climate change. In particular, vegetation responds very differently from P/Rnet or P/PET when CO2 is involved (as far as we can tell.) Another way to see this is that satellite data shows the Earth is greening in recent decades (Zhu et al. 2016 Nature Climate Change) even while a P/PET calculation would tell you it has been drying (some of Fu's papers which analyzed historical data). This implies P/PET is not just "imperfect" for vegetation change under CO2/climate change, but grossly misleading.*

Ans: Thank you for making such a thorough and constructive comment. We were aware that the usefulness of P/PET is limited and as you say we tried to get some qualitative understanding of the problem using this rather blunt tool. However, we agree that relying solely on P/PET risks giving a biased and potentially misleading perspective on our results. We have therefore decided to exclude Section 2.2 and Section 6 based on your comment and the references in it.

Replacing it with direct output from the land model is a good suggestion, but when looking further into this issue, we realized that the land model that we used, CLM2, has CO2 concentration hardcoded in the model and the vegetation will not be affected by changes in CO2 concentration specified for the atmosphere. This means that

stomatal conductance will not be affected by the change in CO2 between the two simulations, biasing the transpiration and P–E values. We have therefore decided not to address terrestrial hydroclimate, refocusing the manuscript towards the precipitation and circulation parts. We have taken up some of the space freed up by adding a new figure and discussion of seasonal and diurnal temperature ranges, in response to your suggestion below. We feel that the resulting manuscript is more solid and defensible, and contains sufficient substance and novelty to warrant publication.

*Minor: (Note, some of these are on the PET parts, so they will not be relevant if you decide to replace the PET analysis as suggested above.)*

*p1 li10: Do you mean 11% greater absolute precip, or 11% larger precip \*response\* to warming? This should be clarified. See p4 li26-29 comment below.*

Ans: We mean that global-mean precipitation in LCTC is 11% greater than in the GHG case. We have clarified this in the abstract and also in the relevant section of page 4, see below.

*pi li24-25: Can you at least include a few words about why one might think the Eocene would have fewer aerosols than the present? I.e., does this hypothesis just come out of nowhere or is there some qualitative speculation/cartoon behind it? Given how vegetated and warm the Eocene was, I would a priori think it would have more aerosols than present, via much-increased biogenic VOC emissions as well as more fires (more fuel.) Though, I suppose it would also have less dust due to the increased vegetation cover. So it could go either way.*

Ans: We have added some additional background and references in the introduction to motivate this assumption, though we emphasize that it remains highly speculative. It is precisely the uncertainty in Eocene aerosol/cloud forcing that serves as the main motivation for this paper.

*p2 li30-31: The atmosphere and ocean models are specified here, but not the land model. What land model was used?*

Ans: The land model used in the simulations was CLM2. That should have been included in the model description. Thank you for pointing this out and we have addressed it in the revised manuscript.

*p2 li32: By prescribing a seasonally-varying q-flux but \*also\* using a slab mixed-layer ocean, you may be double-counting the seasonal storage and release of heat in the mixed layer: : : be careful here. What is the depth of the slab ocean? If you are going to use a seasonally-varying q-flux, your slab ocean should probably have minimal depth, since the seasonal cycle of the q-flux is \*already\* dominated by the fluxes in and out of the ocean every year due to seasonal warming and cooling of the mixed layer. (Does this make sense?) In this case, you should also insert "and seasonal storage" after "transport" on line 31.*

Ans: That is a good point. The qflux used is indeed seasonally varying, and the slab has 50 m uniform depth (this detail has been added in the Methods section). While the annual-mean temperature is in good agreement with the coupled model simulation of

Huber and Caballero (2011) (~2C differences in the zonal- and annual mean), there are larger seasonal differences, which makes the mid/high latitude annual temperature range in both hemispheres of our GHG simulation about 5C smaller than in the coupled model. There is no easy way around this problem. The qflux is determined from the climatological monthly-mean net surface heat flux in the coupled model, which—as the reviewer correctly points out—includes a storage term. The storage term is negligible in the annual mean, but specifying a time-invariant annual-mean qflux would also lead to bias since ocean heat transport does have a seasonal variability. However, determining the actual seasonally-varying ocean heat transport would involve computing it from the 3-dimensional ocean current and temperature fields in the coupled model, to which we do not have access. In any event, our paper focuses on comparing two simulations run with the same qflux, so we believe that our results will be at least qualitatively robust to different specifications of this qflux. We have added some more text in the methods section discussing this issue.

*p3 li25: For the reader unfamiliar with hydrology and PET, you should also mention that capital Delta is the slope des/dT of the es curve (instead of just giving its numerical formula out of context.)*

Ans: This is a good suggestion but it is no longer relevant since we have excluded the P/PET part of the paper.

*p3 li29-32: Scheff and Frierson (2015, the follow up to the 2014 paper) checked the monthly assumption more closely. They found that it's not bad at all for changes in PET (probably because the day-night warming difference isn't actually that big in most places) though it can throw off the absolute PET values by double-digit percentages. However, they only tested greenhouse-driven warming, so I don't know if the result would be valid for LCTC/shortwave warming. Presumably the LCTC world warms much more during the day than at night, so this time-resolution issue could actually become important. In fact, the diurnal cycle of temperature could be another major constraint detectable in the paleoecological record - plant species composition may respond quite differently to warming with much-increased diurnal cycle (LCTC) than warming with unchanged or somewhat reduced diurnal cycle (CO2) because different species have different tolerances to min and max temperature. Again this would be independent of any watercycle- driven vegetation change. Thus, it would be interesting for you to quantify the difference in diurnal Tmax and/or diurnal Tmin between your two Eocene end-members, not just the difference in Tmean. Either of these may be quite dramatic, even though the Tmean doesn't differ much. All of this may additionally hold for seasonal cycle (winter and summer temperatures rather than annual-mean temperature.) We have a good idea of winter temperatures from some of the proxies like crocodiles and palm trees surviving in the Arctic. Some SST proxies are also particularly sensitive to summer temperature so they could be useful here.*

Ans: This is an excellent suggestion, and we have added a new figure (Fig 3 in the revised manuscript) as well as 3 new paragraphs of text in Section 3 discussion changes in seasonal and diurnal temperature ranges.

*General:*
*The first mention of Figure 1 right now (p4 li12) comes after the first mention of*

*Figure 2 (p4 li5). So these two figures should be reversed and re-numbered accordingly. I.e. you need to re-number the current Figure 2 as Figure 1, and vice versa.*

Ans: The two figures in question have been renumbered in the revised manuscript.

*p4 li26-29: Not only is it a substantial increase due to this reasoning, it's also a substantial increase because it's on the same order or larger than what you would get by differencing LCTC with a Holocene run (or differencing high-CO2 with a Holocene run.) In general it might be more useful to contrast the differences of high-CO2 and LCTC with a common Holocene-like (i.e. low-CO2, normal-cloud) control, instead of contrasting the absolute precip values. I.e. by what factor or percentage is the LCTC-Holocene difference larger than the highCO2-Holocene difference. It will be a lot more than 11%. And it's that difference from present that we tend to notice when we think about Earth system change over time, not the absolute value.*

Ans: The precipitation increase from modern to GHG is actually about twice as large as the increase from GHG to LCTC, in agreement with the precipitation scaling implied by the Carmichael et al paper. We have added some text in page 4 to point this out. We agree that including a comparison of both GHG and LCTC cases with a modern preindustrial run would be of interest, but to do such a comparison justice would require a large number of additional figures and a lot more analysis and text. We have added references to papers providing extensive information on modern CAM3 simulations (Hurrell et al. 2006, Collins et al. 2006) that the interested reader may consult.

*p4 li29-34: This needs to cite Fig 2d.*

Ans: We have added the citation in the revised manuscript.

*p5 li3-4: No it doesn't "only" prescribe change in cloud droplet size - it also prescribes a big change in CO2 (of course.) The effects of the CO2 difference could also be important for the big cloud cover changes you see. Again, differencing both simulations with a common low-CO2 normal-cloud control would help in disentangling this.*

Ans: Yes, that is of course true; we meant that the only cloud property directly modified in the LCTC case is droplet size. We have modified the sentence in question to include the change in CO2. The issue of disentangling the separate effects of droplet size and CO2 is addressed by the fixed SST runs, in which these parameters are changed one at a time (which is a cleaner solution, since a modern simulation would have different geography and land cover, adding confusion).

*p5 li10: Not to sound like a broken record here, but does a "modern" (Holocene-like) control simulation with this model also have this Pacific meridional tilt? You may just be seeing a general bias of the model, not a particular effect of the Eocene climate. Exact same issue with "than observed in the modern climate" at li14-15, too. If Eocene model is different from modern observation, it's hard to tell whether that difference is due to "Eocene vs. modern" or whether it's due to "model vs.*

*observation" (without further information or citations.)*

Ans: No, as shown in Hurrell et al. 2006, a modern simulation with CAM3 captures the Pacific jet quite well (and does not have the large meridional tilt seen in our GHG simulation). We have added this reference and some additional text to clarify this point.

*p6 li1: Strictly speaking this equation will give a negative value for P, since LH as you just defined it will be a negative number (LH is upward, not downward) while Lv is positive. So you may want a minus sign in this equation to be technically correct.*

Ans: This is correct and we have added a minus sign in the revised manuscript.

*p6 li8 and Fig 3: It's not immediately clear what bars in Figure 3 to look at to see "Atmospheric heating". It seems for SW and LW it's the black (TOA-surface) bar, but for LH and SH it's actually the orange (surface) bar. So this would be easier to read if you made black bars for LH and SH as well, and explicitly pointed out in the text or the figure caption that the black (TOA-surface) is the (net) atmospheric heating in all cases. In this case you would also flip all of the orange bars to opposite sign, to be consistent with your definition of negative up, positive down (see previous comment.) Then, the coloring would be consistent across all four heat transfer mechanisms. Alternatively you could keep your opposite-sign convention for the orange bars, but then you should rename the black bars to "atmospheric heating" rather than TOA minus surface, since strictly speaking they are TOA plus surface in the current graphical setup.*

Ans: This was unclear in the manuscript and we have changed Figure 3 to make it consistent according with the first suggestion above. In addition, we added a clarification in the caption that the black bars are the atmospheric heating.

*Also p6 li8: it's hardly "dominated" by the LH component, the SW component is actually almost as big! Rather you could just say that LH is the largest component (a lot more defensible.) "Dominated" means much larger than the others, not just larger.*

Ans: This is a good point, we have changed the text accordingly in the revised manuscript.

*p8 li4: I assume this is supposed to be "westerly anomaly" (i.e. east-pointing.)*

Ans: It is corrected in the revised manuscript to westerly anomaly.

*p9 li17-18: This should have a citation (e.g. to Middleton and Thomas) so the reader can know who is doing the classifying.*

Ans: This is not relevant in the revised manuscript.

*p9 li19-20: Low relative humidity is also very important for this, I think (you could check this.)*

Ans: This is not relevant in the revised manuscript.

*p10 li27: As alluded in the major comment, "eco-hydrological differences" is too vague because plants, rivers, soils, precip, etc. all respond to CO2-driven climate change in such a different manner from one another. If you mean vegetation, just say vegetation. (If you mean something else, say something else.)*

Ans: This is not relevant in the revised manuscript.

*Figures 7a and 2c: It's hard to see the precipitation patterns over most of the planet, because most of the map is in the very light colors which don't distinguish from each other easily. You may want to have your color scales saturate to dark at lower precipitation values (e.g. 9 instead of 18) to make the patterns easier to see. More than 9 mm/day is very wet by anyone's measure.*

Ans: Figure 7a is excluded in the revised manuscript. See answer to the major issue above. It is a good suggestion and makes Figure 2c much more informative.

*Figures 7a and 7c: Color values still seem to be plotted over the ocean, even though you are not trying to plot ocean points. This is a little confusing, and for 7a it worsens the above problem. Make sure the oceans are actually white here (or some other neutral color.)*

Ans: Figure 7a and 7c is excluded in the revised manuscript See answer to the major issue above.

*Typos:*
*p2 li30: "research" needs to be capitalized in the name of NCAR.*
*p4 li22: should be "subtropical anticyclones" rather than "subtropical cyclones" (I presume?)*
*p5 li9: "zonal winds are in the Northern Hemisphere: : : are concentrated" should be "zonal winds in the Northern Hemisphere: : : are concentrated"*
*p8 li14: extra word "may" near the beginning of this line*

Ans: All the above typos have been addressed in the revised manuscript

[revised manuscript text omitted]